# Diversity of Gut Microbiota Affecting Serum Level of Undercarboxylated Osteocalcin in Patients with Crohn’s Disease

**DOI:** 10.3390/nu11071541

**Published:** 2019-07-08

**Authors:** Kohei Wagatsuma, Satoshi Yamada, Misora Ao, Minoru Matsuura, Hidemi Tsuji, Tomoya Iida, Kentaro Miyamoto, Kentaro Oka, Motomichi Takahashi, Kiyoshi Tanaka, Hiroshi Nakase

**Affiliations:** 1Department of Gastroenterology and Hepatology, Sapporo Medical University School of Medicine, Sapporo 060-8543, Japan; 2Department of Gastroenterology and Hepatology, Kyoto University Graduate School of Medicine, Kyoto 606-8507, Japan; 3Department of Food and Nutrition, Kyoto Women’s University, Kyoto 605-8501, Japan; 4Faculty of Home Economics, Kobe Women’s University, Kobe 654-8585, Japan; 5Tokyo R&D Center, Miyarisan Pharmaceutical Co., Ltd., Tokyo 114-0016, Japan; 6Faculty of Nutrition, Kobe Gakuin University, Kobe 651-2180, Japan

**Keywords:** vitamin K, undercarboxylated osteocalcin, gut microbiota, Crohn’s disease, diversity

## Abstract

Several reports have indicated a possible link between decreasing plasma levels of vitamin K and bone mineral density. It has been suggested that intestinal bacteria contribute to maintenance of vitamin K. Several factors are involved in the reduction of vitamin K in patients with Crohn’s disease (CD). We aimed to assess the relationship between gut microbiota and alternative indicators of vitamin K deficiency in patients with CD. We collected the feces of 26 patients with clinically inactive CD. We extracted 16S rRNA from the intestinal bacteria in the feces and amplified it by polymerase chain reaction. The generated polymerase chain reaction product was analyzed using a 16S metagenomic approach by Illumina Miseq platform. Serum undercarboxylated osteocalcin concentration was used as an alternative indicator of vitamin K deficiency. There was a significant negative correlation between serum undercarboxylated osteocalcin and mean Chao1 index in cases of low activity. The diversity of the gut microbiota was significantly lower, and *Ruminococcaceae* and *Lachnospiraceae* were significantly decreased in the vitamin K-deficient group in comparison to the vitamin K-normal group. Taken together, these data suggested the significance of investigating the gut microbiota even in patients with clinically inactive CD for improving patients’ vitamin K status.

## 1. Introduction

Vitamin K (vit K), a fat-soluble vitamin, exists in some forms: vit K1 (phylloquinone), which is mainly found in green leafy vegetables, and vit K2 (menaquinone-n: MK-n), which is mainly found in fermented dairy and produced by lactic acid bacteria in the intestine [1,2]. In addition, vit K3 (menadione) is a synthetic vit K, which is metabolized in the animal to vit K2. Vit K3 is added to animal feed as a vit K source. Vit K2 is classified by the number of isoprene units in the side chain. MK-4 is relatively abundant in chicken meat and egg. Natto, which is a Japanese fermented soybean-based product, is rich in MK-7, also contains MK-8. Several types of cheese contain MK-8 and MK-9 [3]. The vit K2 synthesized by intestinal bacteria are mainly MK-10 and MK-11, but small amounts of MK-7, MK-8, MK-9 and MK-12 are also synthesized [4]. In particular, vit K2 plays an important role in regulating the activity of matrix Gla protein and osteocalcin after being transported to extrahepatic tissues. In addition, there are Vit K-dependent proteins that are carboxylated by vit K in places like periosteum and periodontal membrane (periostin), brain and vascular endothelial tissues (growth arrest specific-6), liver (transthyretin), cornea (β-Ig-H3), biomembrane (PRGP-1, 2; TMGP-3, 4) and so on. Vit K promotes blood coagulation [5,6], bone fracture healing, and osteogenesis [7,8]. Also, vit K is reported to be related to various organs and diseases such as the chronic kidney disease, some types of cancer, liver disease, immunity functions, neurological disorders, obesity and so on. Vit K1 is preferentially retained in the liver to aid the carboxylation of coagulation factors [9]. In contrast, vit K2 is redistributed into the circulation and is used for extrahepatic tissues such as bones and the vascular system [9]. The mechanism of clinical outcomes can be complicated because the work of vit K varies widely. Several reports have indicated a possible link between decreasing plasma levels of vit K and bone mineral density [10]. Supplementation of phylloquinone had modestly increased the bone mass of the total radius [11]. It has been reported that increased dietary intake of vit K may reduce the risk of fractures [12]. On the other hand, the association between vit K and the risk of fractures is still debatable because there were few well-designed prospective cohort studies and clinical trials. Many broad-spectrum antibiotics decrease the number of intestinal bacteria, limit vit K production, and cause coagulopathies [13,14,15]. Vit K is known to be deficient in newborn babies whose intestinal bacteria are not well established, and asymptomatic mice raised with food lacking vit K have an increased mortality rate with deterioration of hemostatic function [16,17]. The germ-free mice exhibited increased bone mass, and a normalization of bone mass after a gut microbiota transplantation [18]. Based on these results, it has been suggested that intestinal bacteria contribute to maintenance of vit K.

Vit K gets absorbed by the enterocytes of the small intestine and is dependent on bile, pancreatic enzymes and the dietary fat content. Vit K2 in human intestines is related to the composition of the gut microbiota but is reported to be highly variable [19]. Also, while vit K is absorbed in the small intestine through a mechanism that requires bile acid salts, most is produced in the colon where there are no bile acid salts. The percentage of vit K synthesized by intestinal bacterial is much lower than excepted previously, although the exact proportion of vit K production/absorption in colon and small intestine remain unclear [20]. However, due to the difference of bioavailability, bioactivity and several unique functions of vit K2 [3,21], it is considered that even small amounts of vit K2 derived from intestinal bacteria can have a significant impact on health. Vit K deficiency is rare in healthy individuals [22], but has been found to be more common in patients with inflammatory bowel disease (IBD) [10,23,24]. It has been considered that malabsorption disorders due to small intestine lesions and/or resections [10,23,25], fat restriction, and the use of glucocorticoids [26,27] could lead to vit K deficiency in patients with Crohn’s disease (CD).

It is well-known that the gut microbiota of patients with CD differs in proportion and composition to that of patients without CD [28]. Furthermore, recent reports have suggested that there may be an association between gut microbiota and osteoporosis [29]. However, there have been no studies on the association between gut microbiota and vit K in patients with CD.

There are many reports indicating that the bones are more sensitive to vit K deficiency than the liver [10,30]. Although vit K γ-carboxylates osteocalcin, in the cases of vit K deficiency, undercarboxylated osteocalcin (uc-OC) is released into the blood [31]. There are two main substrates in vit K dependent carboxylation that are used to assess vit K status. These two substrates are uc-OC and protein induced by vit K absence-II (PIVKA-II), because direct measurement of vit K is not easy in routine practice. Increasing serum levels of uc-OC and PIVKA-II reflect vit K deficiency in the bone and liver, respectively. The increase of serum uc-OC indicates vit K deficiency. Dietary vit K intake for healthy women affects serum levels of uc-OC [32]. The serum levels of uc-OC negatively correlated with the vit K intake in anorexia nervosa patients showing bone loss [33]. Elevated serum level of uc-OC indicates the increased risks of hip fractures in elderly women [34,35]. It was reported that the serum levels of uc-OC were more strongly related to ultrasound transmission speed than to femoral neck density [36]. An observational study on adult patients with CD, demonstrated that the serum levels of uc-OC positively correlated with bone turnover speed [37] and to negative correlate with BMD in the lumbar spine [23].

The aim of this study was to assess the relationship between the gut microbiota and alternative indicators of vit K deficiency in patients with CD.

## 2. Materials and Methods

### 2.1. Ethical Statement

This study was approved by the institutional review board of Kyoto University (E2315, December 24, 2014). Informed consent was obtained from all individual participants included in the study before enrolling. All procedures performed in studies involving human participants were in accordance with the ethical standards of the institutional and/or national research committee and with the 1964 Helsinki declaration and its later amendments or comparable ethical standards.

### 2.2. Study Participants

Eligibility criteria were outpatients with CD treated at Kyoto University Hospital, who agreed to participate in this study between August and December 2015. CD was diagnosed based on clinical diagnostic criteria by symptoms, radiological findings, endoscopic findings, and histological findings. Exclusion criteria were (1) patients with clinically active disease; (2) patients that were pregnant or are likely to be pregnant; (3) patients who did not agree with the epidemiological study; and (4) other cases deemed inappropriate by the attending physician or the conducting physician. There were 26 patients with CD from whom we obtained consent and from whom we were able to collect feces samples.

### 2.3. Analysis of Gut Microbiota (Meta-16S rRNA Gene Sequence Analysis)

The collected amount of feces was 5–10 g and the feces were stored at −80 °C until measurement. The feces were immediately discarded after measurement. DNA was extracted from the collected feces using a glass bead extraction method and purified according to a previously reported method [38]. The V3–V4 region of the 16S rRNA gene was amplified via polymerase chain reaction (PCR) using barcoded primers, and PCR products were sequenced using the paired-end technique (Illumina MiSeq platform), as previously described [39]. By using the quantitative insights into microbial ecology (QIIME 1.8.0) [40], 966,729 high-quality reads were generated and assigned to operational taxonomic units (OTUs). A representative sequence for each OTU was aligned with Python Nearest Alignment Space Termination [41], and taxonomically classified by using UCLUST [42].

### 2.4. Measurement of Vit K

In our study, the serum uc-OC concentration was used as an alternative indicator. Regarding the cut off value for serum level of uc-OC, the cut-off value for serum level of uc-OC in osteoporosis is set at 4.5 ng/mL at many institutions in Japan based on the previous data in Japan [43]. The patients were divided into two groups based on the cutoff level of the uc-OC concentration, and the gut microbiota were compared.

### 2.5. Endpoints and Examination Items

The primary endpoint was indicated by any association between the gut microbiota and uc-OC in the patients with CD. The secondary endpoint was a clarification of factors related to uc-OC decline. Other examination items that were evaluated included: age, duration of disease, body mass index, clinical disease type (inflammatory, stricturing, penetrating), disease location range (small intestine, small and large intestine, large intestine), presence of anal lesions, surgical history (small bowel resection, ileocecal resection), disease activity (Crohn’s disease activity index: CDAI), blood test results (albumin, total cholesterol, triglyceride, cholinesterase, calcium, C-reactive protein, intact parathyroid hormone, uc-OC, PIVKA-II, folic acid, vitamin B12, and homocysteine), and therapeutic regimen (antibiotics: cefepime/metronidazole/clarithromycin, probiotics, immunosuppressants, biologics). Ten mL of whole bloods were collected for research purposes for intact parathyroid hormone, uc-OC, PIVKA-II, folic acid, vitamin B12, and homocysteine measurements. The serum samples were stored at −20 °C until measurement and was immediately discarded after measurement. Other blood test items were measured at blood collection volume within the range of daily medical practice. PIVKA-II and uc-OC levels were measured by electrochemiluminescent immunoassay (ECLIA).

### 2.6. Statistical Considerations

Normally distributed continuous variables were analyzed using the Student’s *t*-test and nonparametric data were analyzed using the Mann–Whitney U test. Categorical variables were analyzed using Pearson’s chi-squared test or Fisher’s exact test if any cell number was less than five. Two-sided *p* levels less than 0.05 were considered statistically significant. All analyses were performed using SPSS software (IBM corp. Armonk, NY, USA).

To analyze diversity of gut microbiota, the alpha diversity and beta diversity were calculated. Alpha diversity at the OTU level (Chao1 index) was calculated in QIIME [44]. Unweighted UniFrac distance was applied for principal coordinates analysis (PCoA) at the OTU level to analyze beta diversity [45], and each group was compared with PERMANOVA. The enriched bacteria in each group were identified by linear discriminant analysis (LDA) effect size (LEfSe) [46]; LDA values >2 were considered significant. Abundant taxa were highlighted on the phylogenetic tree using the GraPhlAn software.

## 3. Results

We investigated the difference in the gut microbiota between patients with a normal uc-OC (<4.5 ng/mL: vit K-normal group) and those with a high uc-OC (≥4.5 ng/mL: vit K-deficient group). The backgrounds of the patients in the two groups based on uc-OC are shown in Table 1. PIVKA-II was significantly higher in the vit K-deficient group than in the vit K-normal group (*p* = 0.01), but there was no significant difference in indicator of nutritional status (such as albumin, total cholesterol, triglyceride, cholinesterase, folic acid, vitamin B12, and homocysteine), disease activity, surgical history, or treatment regimens, including the use of antibiotics. Regarding the use of warfarin and methotrexate, there were no patients receiving these drugs in this study.

To determine alpha diversity, we calculated the mean Chao1 index (Figure 1). The mean Chao1 index was significantly lower in vit K-deficient group than in vit K-normal group (*p* = 0.0044). There was also a significant negative correlation between uc-OC concentration and mean Chao1 index in cases of low activity (CDAI <150) (Figure 2).

To determine beta diversity, the unweighted UniFrac distances were applied for PCoA at the OTU level (Figure 3). Comparison between vit K-deficient group and vit K-normal group with PERMANOVA showed a significant difference in unweighted UniFrac distances (*p* = 0.013).

We analyzed taxonomic comparison in gut microbiota between vit K-deficient group and vit K-normal group at family and genus levels. Gut microbiota at the family level in the two groups in the vit K-satisfactory state is shown in Figure 4. We applied LDA combined LEfSe to explore the enriched gut microbiota in the vit K-deficient group and vit K-normal group at family level (Figure 5). Abundant gut microbiota at the family level were highlighted on the phylogenetic tree using the GraPhlAn software (Figure 6). Family *Enterococcaceae*, family *Paraprevotellaceae*, and families in order *Clostridiales* were significantly increased in the vit K-deficient group. In contrast, families in division *Firmicutes*, family *Lachnospiaceae*, and family *Ruminococcaceae* were significantly decreased.

Gut microbiota at the genus level in the two groups in the vit K-satisfactory state is shown in Figure 7. We applied LDA combined LEfSe to explore the enriched gut microbiota in the in vit K-deficient group and vit K-normal group at genus level (Figure 8). Abundant gut microbiota at genus level were highlighted on the phylogenetic tree using the GraPhlAn software (Figure 9). At the genus level in vit K-deficient group, genus *Enterococcus*, genera in *Enterococcaceae* family, and genera in order *Clostridiales* were significantly increased in the vit K-deficient group. In contrast, genus *Blautia*, genera in family *Lachnospiraceae*, genus *Anaerotruncus*, genus *Ruminococcus* in family *Lachnospiraceae*, genus *Dorea*, genera in family *Erysipelotrichaceae*, genus *Oscillospira*, genus *Ruminococcus* in family *Ruminococcaceae* were significantly decreased.

## 4. Discussion

To the best of our knowledge, this study was the first to demonstrate the association between gut microbiota and uc-OC, which is an alternative indicator of vit K deficiency, in patients with CD. We found that the diversity of gut microbiota decreased in vit K-deficient group compared to vit K-normal group. Also, there was a significantly negative correlation between serum concentration of uc-OC and Chao1 index. Of note, taxonomic analysis demonstrated that the composition of gut microbiota in vit K-deficient group of patients with inactive CD tended to mimic that of patients with active CD. Taken together, these data suggested the significance of investigating the gut microbiota even in patients with clinically inactive CD for improving patients’ nutritional status.

The number of IBD patients has been globally increasing [47]. A growing body of evidence links the pathophysiology of IBD with the altered microbiota composition [48,49]. Therefore, gut microbiota are considered to play a crucial role in the etiology of IBD. Recently, it was reported that most microbiota changes seem to occur early in the disease course and may be both causal and responsive. There is a huge amount of interindividual heterogeneity and (in the Human Microbiome Project 2 data) little influence on disease progression, behavior, and disease activity measurements [50].

In contrast, it is acknowledged that the gut microbiota have the capacity to synthesize a variety of vitamins involved in a myriad of aspects of microbial and host metabolism. Previously, we reported the association between plasma vit K and 25-hydroxyvitamin D concentrations and bone mineral density (BMD) in IBD patients [10]. Data suggested that patients with CD had significantly lower plasma vit K, higher serum level of uc-OC, and lower BMD scores at almost all measurement sites compared to patients with ulcerative colitis. More IBD patients were vit K-deficient in bone than liver. Multiple regression analyses revealed that low plasma concentrations of vit K were independent risk factors for low BMD. Based on these data, we performed further investigation to examine how the gut microbiota in CD patients affect vit K by using uc-OC as an alternative indicator.

First, we classified the enrolled patients into two groups based on the uc-OC level. There was no significant difference that would cause a decrease in vit K, such as surgical history, disease activity, treatment regimens, and use of antibiotics between the vit K-deficient group and the vit K-normal group. To determine alpha diversity, we calculated the mean Chao1 index. The diversity of the gut microbiota decreased in vit K-deficient group because the mean Chao1 index was significantly lower in vit K-deficient group than in vit K-normal group. In addition, to determine beta diversity, the unweighted UniFrac distances were applied for PCoA at the OTU level. Comparison between vit K-deficient group and vit K-normal group with PERMANOVA showed a significant difference in unweighted UniFrac distances. These results indicate that the types of bacteria were significantly different between the vit K-deficient and vit K-normal groups. In addition, we found that there was a significantly negative correlation between uc-OC concentration and Chao1 index in patients with clinically inactive CD. Thus, our study suggested that a decrease in diversity of the gut microbiota could lead to a decrease in vit K production. As for the nutritional condition, enrolled CD patients in this study had maintained clinical inactive with keeping good nutritional status in laboratory data. Of note, our data strongly indicates the increase of uc-OC even in CD patients with good nutritional condition, which is an important and informative result for physicians.

Next, we analyzed taxonomic comparison in gut microbiota between vit K-deficient and vit K-normal groups at family and genus levels. We found that *Enterococcaceae*, which is a type of lactic acid bacteria that produces vit K, was abundant in the vit K-deficient group. This data suggested that production of vit K did not depend on *Enterococcaceae* alone. In addition, in the vit K-deficient group, we also found a significant decrease of *Firmicutes*, especially bacteria producing short chain fatty acids, such as *Ruminococcaceae* and *Lachnospiraceae*. Short chain fatty acids include anti-inflammatory molecules, such as butyric acid, and it has been suggested that the decrease of short chain fatty acids leads to the onset and persistence of intestinal inflammation [51]. We found that the gut microbiota in the vit K-deficient group of patients with CD were disturbed despite the patients being clinically inactive. Several reports indicated that patients with active CD have lower diversity of microbiota and a decreased abundance in bacteria producing butyric acid [52,53,54]. Interestingly, our study revealed that the composition of gut microbiota of the vit K-deficient group in patients with inactive CD had a similar tendency to that of patients with active CD. This finding may suggest the significance of gut microbiota analysis in the total management of patients with CD.

Recent studies using germ-free mice and probiotics have demonstrated the influence of the gut microbiota in regulating bone physiology [18,55], although the exact role that the microbiota play in the development of bone is complicated. The study of the gut microbiota in patients with osteoporosis by Wang, et al. showed a high ratio of *Firmicutes*/*Bacteroidetes* divisions at the division level and high proportions of genus *Ruminococcaceae* in osteoporosis patients [29]. In their study, they did not examine vit K level in enrolled patients; therefore, we could not directly compare our taxonomic data to theirs. However, the composition of gut microbiota in vit K-deficient group was different from that reported by Wang, et al. This finding suggested, based on microbiota analysis, that the mechanism of decreasing BMD in patients with CD might be different from that of osteoporosis, which was not related to IBD.

There are several limitations in our study. First, we enrolled patients with clinically inactive Crohn’s disease at a single medical institution. There were no specific eligible criteria for the enrollment. Only a small number of cases were enrolled in this study. Therefore, we could not exclude the bias regarding the effect of several therapies such as antibiotics, probiotics, and immunosuppressants, which would disrupt the diversity of gut microbiota in CD patients. Second, we did not have the exact information of diet, which would affect gut microbiota. Third, using % uc-OC with more indicators may be ideal for the evaluation of vit K deficiency [56]. However, in Japan, the measurement of uc-OC is approved for osteoporosis, while those of vit K and OC are not covered by Japan government insurance. Therefore, we evaluated vit K condition only by serum level of uc-OC, which is an available parameter in daily clinical practices in Japan. Fourth we did not evaluate the association between vitamin D (vit D) and microbiota in this study. It is well recognized that the intestinal inflammation of CD results in the deficiency of vit K and vit D. Also, vit D is strongly associated with osteogenesis. Therefore, further investigation will be required the association between the metabolism of vit D and microbiota in quiescent CD patients. But, the relationship of uc-OC with vit D and bone markers has been considered in a previous report [10], so this time we focused on vit K, which is closely related to intestinal bacteria. Fifth, the causal relationship between the gut microbiota and the uc-OC concentration remains unclear because this was a cross-sectional study. Therefore, it will be necessary to examine whether administration of vit K could change the gut microbiota, and whether manipulation of gut flora with probiotics, prebiotics administration, and fecal transplantation could change vit K level. Moreover, we need to clarify whether such interventions could lead to the improvement of bone mineral density. Additionally, we might consider the vit k administration affecting the reduction of intestinal inflammation in patients with CD and the increase of nutrient absorption because of its anti-inflammatory properties [57].

## 5. Conclusions

The gut microbiota in the vit K-deficient group of patients with clinically inactive CD showed a similar tendency to that in patients with highly active CD as seen in previously reported literature. This was different from the tendency in patients with osteoporosis but without IBD. These data suggested the significance of investigating the gut microbiota even in patients with clinically inactive CD for improving patients’ vit K status.

## Figures and Tables

**Figure 1 nutrients-11-01541-f001:**
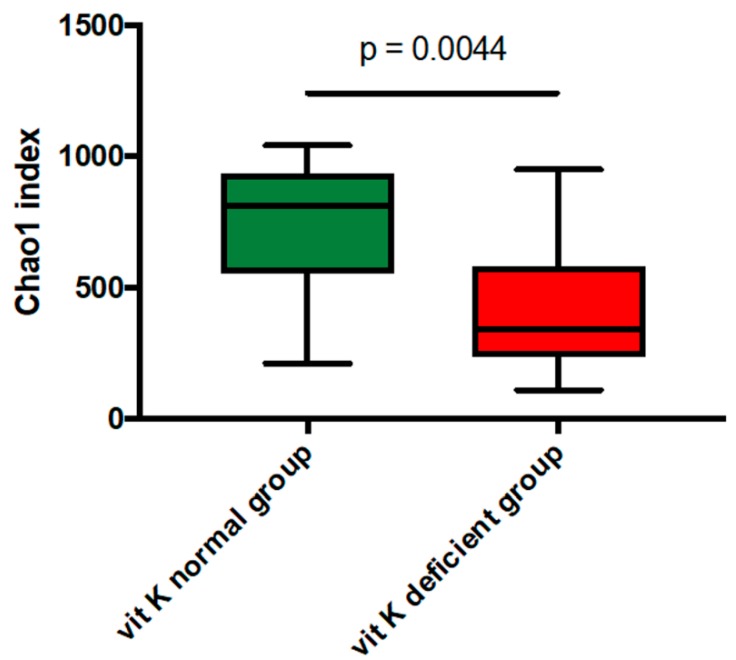
The mean Chao1 index between vitamin K-deficient group and vitamin K-normal group. The mean Chao1 index was significantly lower in vitamin K-deficient group than in vitamin K-normal group (*p* = 0.0044). vit K, vitamin K.

**Figure 2 nutrients-11-01541-f002:**
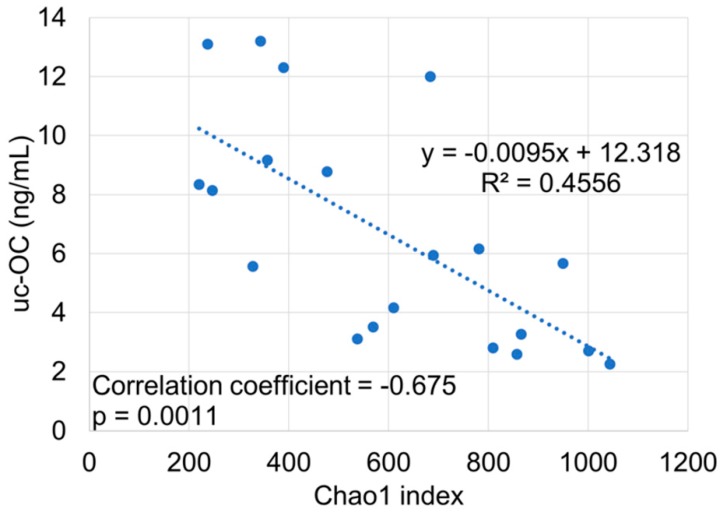
Correlation between undercarboxylated osteocalcin concentration and mean Chao1 index in cases of low activity. There was also a significant negative correlation between undercarboxylated osteocalcin concentration and mean Chao1 index in cases of low activity (Crohn’s Disease Activity Index: CDAI <150). uc-OC, undercarboxylated osteocalcin.

**Figure 3 nutrients-11-01541-f003:**
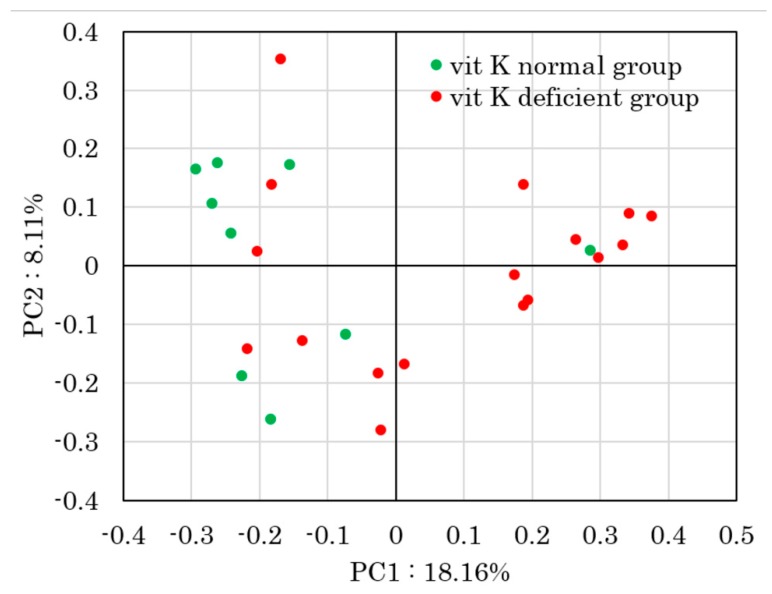
Principal coordinates analysis (PCoA) derived from unweighted UniFrac distances. Comparison between vitamin K-deficient group and vitamin K-normal group with PERMANOVA showed a significant difference in unweighted UniFrac distances (*p* = 0.013). vit K, vitamin K; PC, principal coordinate.

**Figure 4 nutrients-11-01541-f004:**
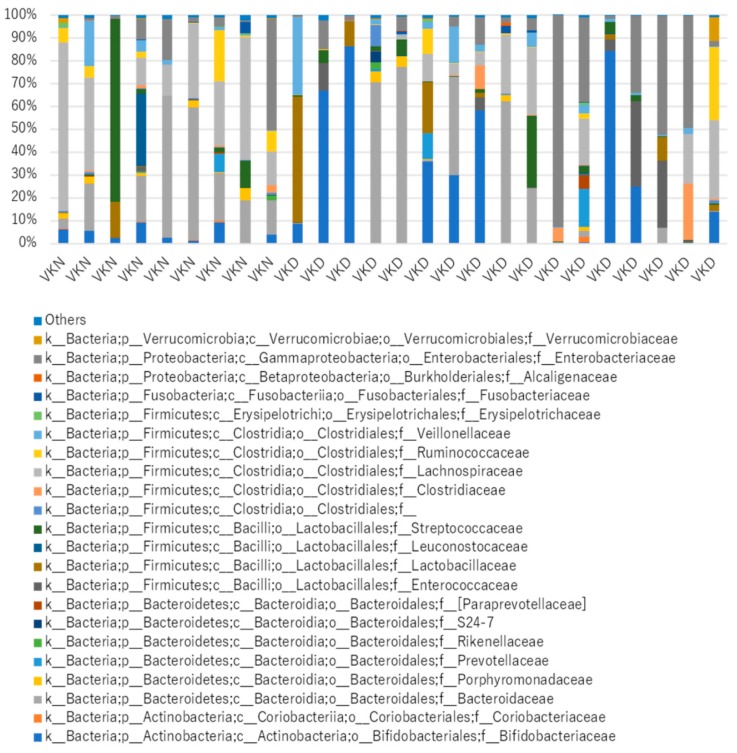
Gut microbiota at the family level in the two groups in 26 patients. VKN, vitamin K-normal group; VKD, vitamin K-deficient group; k_, kingdom; p_, phylum; c_, class; o_, order; f_, family.

**Figure 5 nutrients-11-01541-f005:**
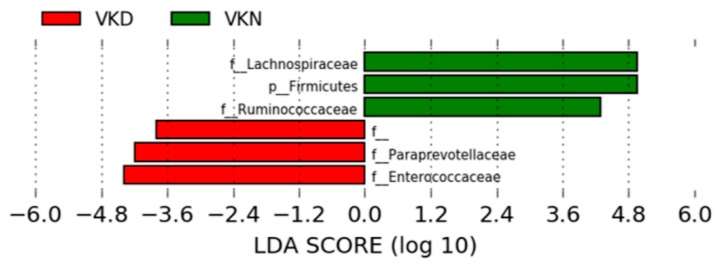
Gut microbiota at the family level in the vitamin K-satisfactory state. The enriched gut microbiota at the family level were identified by linear discriminant analysis (LDA)effect size (LEfSe). The operational taxonomic units (OTUs) with an LDA score of >2 are shown. VKN, vitamin K-normal group; VKD, vitamin K-deficient group; LDA, linear discriminant analysis; f__, order__*Clostridiales*;family__; p, phylum; f, family.

**Figure 6 nutrients-11-01541-f006:**
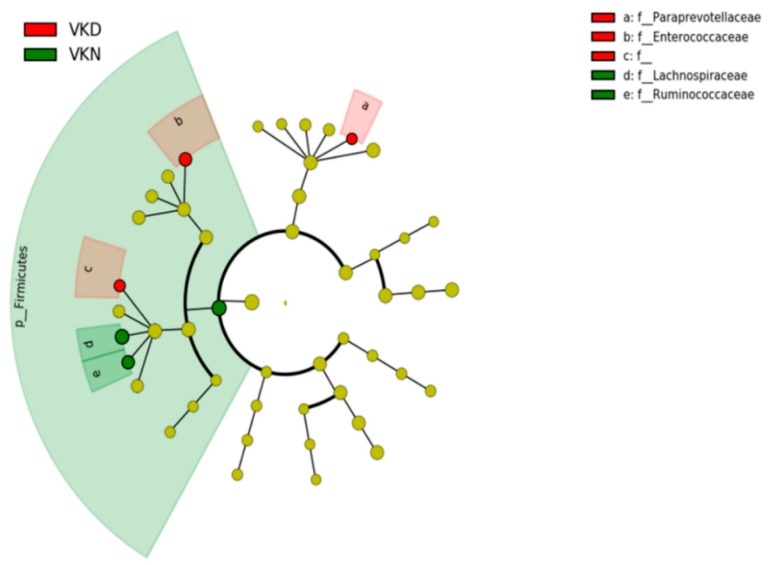
Gut microbiota at the family level in the vitamin K-satisfactory state. Abundant gut microbiota at the family level were highlighted on the phylogenetic tree using the GraPhlAn software. Colors distinguish between vitamin K-normal group (green) and vitamin K-deficient group (red), and the intensity reflects the linear discriminant analysis (LDA) score. The size of the nodes correlates with their relative and logarithmically scaled abundances. VKN, vitamin K-normal group; VKD, vitamin K-deficient group; c: f__, c: order__*Clostridiales*; family__; f_, family.

**Figure 7 nutrients-11-01541-f007:**
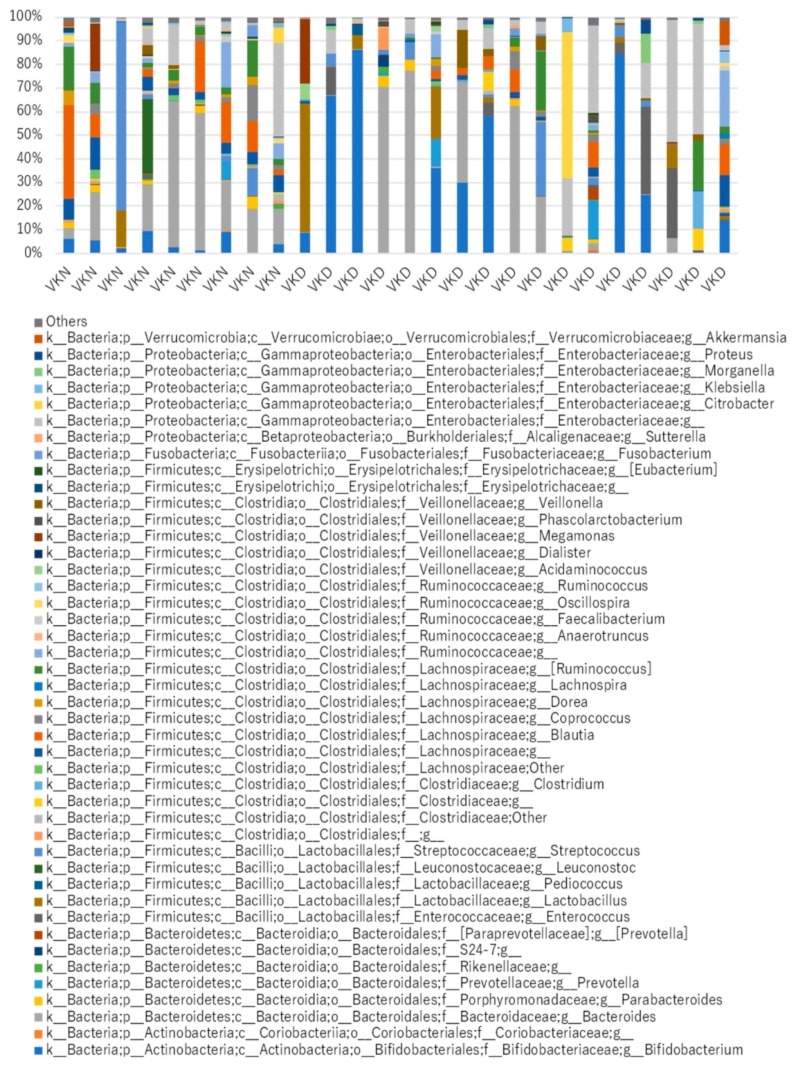
Gut microbiota at the genus level in 26 patients. VKN, vitamin K-normal group; VKD, vitamin K-deficient group; k_, kingdom; p_, phylum; c_, class; o_, order; f_, family; g_, genus.

**Figure 8 nutrients-11-01541-f008:**
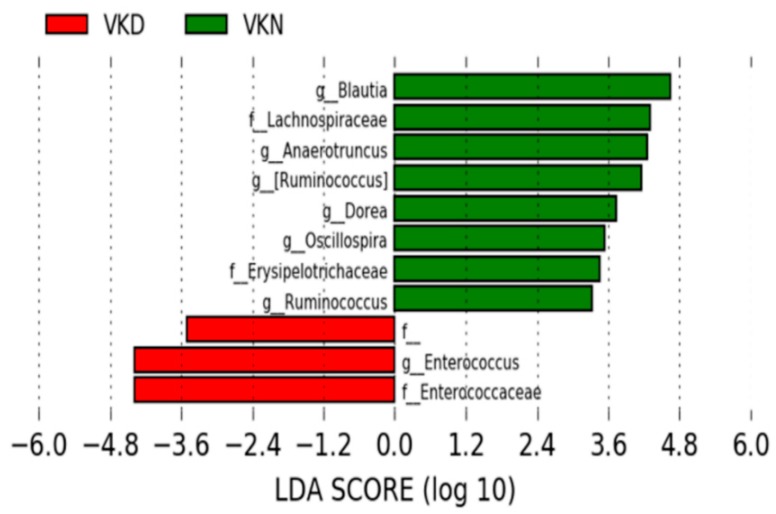
Gut microbiota at the genus level in the vitamin K-satisfactory state. The enriched gut microbiota at the genus level were identified by linear discriminant analysis (LDA) effect size (LEfSe). The operational taxonomic units (OTUs) with an LDA score of >2 are shown. VKN, vitamin K-normal group; VKD, vitamin K-deficient group; LDA, linear discriminant analysis; f__, order__*Clostridiales*;familty__; f, family; g, genus.

**Figure 9 nutrients-11-01541-f009:**
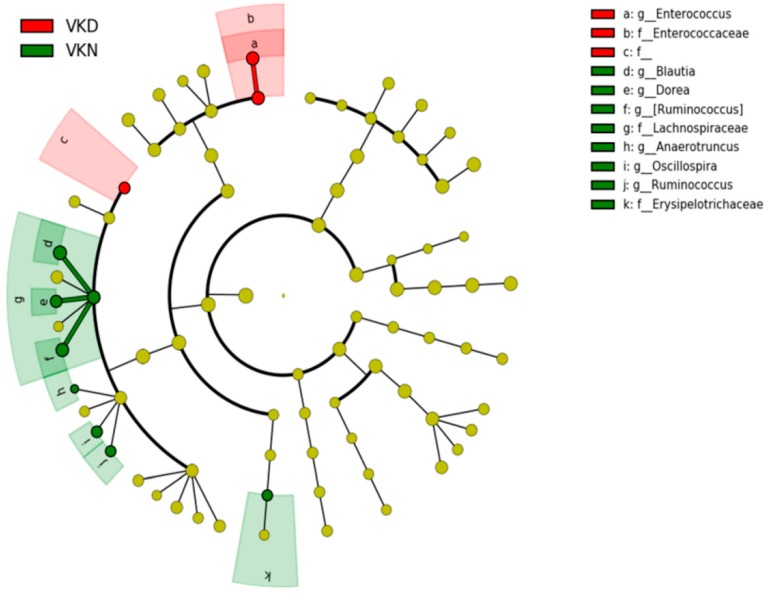
Gut microbiota at the genus level in the vitamin K-satisfactory state. Abundant gut microbiota at the genus level were highlighted on the phylogenetic tree using the GraPhlAn software. Colors distinguish between vitamin K-normal group (green) and vitamin K-deficient group (red), and the intensity reflects the linear discriminant analysis (LDA) score. The size of the nodes correlates with their relative and logarithmically scaled abundances. VKN, vitamin K-normal group; VKD, vitamin K-deficient group; c: f__, c: order__*Clostridiales*; family__; f_, family; g_, genus.

**Table 1 nutrients-11-01541-t001:** Characteristics of the patients for the two groups, based on levels of undercarboxylated osteocalcin.

		uc-OC <4.5 ng/mL (Vit K-normal Group) (*n* = 9)	uc-OC ≥4.5 ng/mL (Vit K-deficient Group) (*n* = 17)	*p*-Value
Age (years; mean, range)		42.8 (28–59)	40.5 (21–73)	0.61
Duration of disease (years; mean, range)		23.0 (11–34)	19.5 (4–53)	0.41
Body mass index (kg/m^2^; mean, range)		22.4 (15.7–27.8)	20.8 (17.8–29.3)	0.32
Clinical disease type	inflammatory/structuring/penetrating	1/7/1	6/6/5	0.19
Disease location range	small/small and large/large intestine	4/5/0	8/6/3	0.75
Anal lesion		3 (33.3%)	6 (35.3%)	0.92
Surgical history	small intestine resection	3 (33.3%)	4 (23.5%)	0.59
	ileocecal resection	3 (33.3%)	8 (47.1%)	0.50
Disease activity (CDAI; mean, range)		90.7 (−9–208)	111.0 (34–204)	0.45
Blood tests	Albumin (g/dL; mean, range)	4.0 (3.0–4.6)	3.9 (3.0–4.1)	0.69
	Total cholesterol (mg/dL; mean, range)	155 (109–209)	141 (97–220)	0.36
	Triglyceride (mg/dL; mean, range)	136 (49–271)	97 (35–166)	0.18
	Cholinesterase (U/L; mean, range)	315 (236–360)	293 (172–497)	0.33
	Calcium (mg/dL; mean, range)	8.9 (8.3–9.2)	8.7 (8.4–9.1)	0.39
	C-reactive protein (mg/dL; mean, range)	0.3 (0–0.9)	0.2 (0–2.3)	0.91
	Intact parathyroid hormone (pg/mL; mean, range)	39.2 (20.0–61.3)	46.7 (15.9–98.5)	0.30
	folic acid	12.3 (4.4–49.3)	6.0 (0.9–10.5)	0.22
	vitamin B12	412 (151–774)	440 (172–853)	0.74
	homocysteine	15.3 (9.1–30.6)	20.7 (9.6–76.8)	0.24
	PIVKA-II (mAU/mL; mean, range)	22.6 (12–32)	31.1 (20–48)	0.01
	uc-OC (ng/mL; mean, range)	3.02 (2.26–4.17)	9.48 (5.56–13.1)	<0.001
Therapy	Antibiotics	2 (22.2%)	9 (52.9%)	0.13
	Probiotics	3 (33.3%)	8 (47.1%)	0.50
	Immunosuppressants	7 (77.8%)	11 (64.7%)	0.49
	Biologics	1 (11.1%)	9 (52.9%)	0.08

CDAI, Crohn’s Disease Activity Index; uc-OC, undercarboxylated osteocalcin; vit K, vitamin K; PIVKA-II, protein induced by vit K absence-II.

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
