# Peer review of "Diversity of Gut Microbiota Affecting Serum Level of Undercarboxylated Osteocalcin in Patients with Crohn’s Disease"

_nutrients, 2019, doi:10.3390/nu11071541_

Reviewer 1 Report

This is a well written research paper outlining the novel role of the gut microbiota on carboxylation status of osteocalcin and thus vitamin K status in remissive Crohn's disease patients. Despite this, there are major study design issues which preclude the reader from really answering the research question at hand and from the authors achieving their research objective.

Introduction is sufficient with a requirement for the inclusion of more relevant studies in support of reduced plasma levels of vitamin K markers and bone density (specifically, L44-45 states that 'several reports' have outlined this yet only 1 ref is provided. Addition of O'Connor et al. 2014 specifically recommended.

No detail is provided on how study participants were selected for enrolment nor is any justification given for the very limited sample size. These are extremely important factors, how did the authors decide on n=26? Was this powered, if so, based on what? It appears too limited and given the fact that it is further sub-divided into two groups (n=9 and n=17) this further reduces the strength of any findings reported thus the reader cannot be confident that the findings are nothing more than spurious.

No exclusion criteria were applied; why? This is surprising as there are several important factors that would need to be considered to ensure the study cohort are, as described, in clinical remission and also not taking medication/therapies that may impact the outcome measures of interest. It is also highly unusual for CD in particular where disease pathology is variable (i.e. differences in terms of site of disease, duration and course of medication etc.) making standardisation (using specific criteria) essential. Specifically, patients should be screened based on medication usage and those taking steroids, methotrexate and warfarin should be excluded. Those taking antibiotics should also be excluded due to their impact on the gut microbiota. Probiotic use (also potentially modulatory to the gut microbiota) was not excluded.

The rationale for use of the 4.5 ng/ml cut point for ucOC is not justified. The authors site one paper for use of this cut-point yet ucOC values are veriable even within apparently similar populations. Furthermore, reporting ucOC as a stand-alone measure of vitamin K status is of limited value. As recommended by Gundberg et al. (1998) it is not accurate to report just ucOC (as an absolute value) without considering total osteocalcin available (which is variable between people). Thus %ucOC should be expressed (considering both cOC and ucOC) when considering this biomarker of vitamin K status. This is not the case in the present study and is a major limitation.

Triglycerides are also not included in the variables reported – this has an impact on vitamin K absorptive capacity and may lead to differences in ucOC levels between patients.

The conclusion suggests this study proves cause and effect of gut microbiota variability on vitamin K status. This is not the case and it is premature to suggest this based on the limited evidence provided here.

Thus the findings reported in this study cannot be considered justified in light of the major limitations outlined above.

Thus the findings reported in this study cannot be considered justified in light of the major limitations outlined above.

Author Response

Response to Reviewer 1 Comments

Point 1: This is a well written research paper outlining the novel role of the gut microbiota on carboxylation status of osteocalcin and thus vitamin K status in remissive Crohn's disease patients. Despite this, there are major study design issues which preclude the reader from really answering the research question at hand and from the authors achieving their research objective.

Response 1: Thank you very much for the kind comment on our manuscript. Also, we really understand what you want to tell us, however, this study is not clinical trial but observational study. Several factors, which might affect metabolism of vitamin K, were not matched between the vitamin K normal group and the deficiency group. Because the enrolled patients at a single medical institution was is relatively small and we recognize it as the limitation of this study. In this study, we investigated the relationship between microbiota and vitamin K status based on the CD patients’ conditions in the real world, therefore, we did not decide the exact eligible criteria. Nevertheless, the significance of our observational study is to demonstrate the interaction between diversity of microbiota and vitamin K status based on the CD patients’ conditions in the real world.

Point 2: Introduction is sufficient with a requirement for the inclusion of more relevant studies in support of reduced plasma levels of vitamin K markers and bone density (specifically, L44-45 states that 'several reports' have outlined this yet only 1 ref is provided. Addition of O'Connor et al. 2014 specifically recommended.

Response 2: Thank you for your great comment. According to your comment, we added some reference in the part of Introduction section in the 65th line of page 2.

Point 3: No detail is provided on how study participants were selected for enrolment nor is any justification given for the very limited sample size. These are extremely important factors, how did the authors decide on n=26? Was this powered, if so, based on what? It appears too limited and given the fact that it is further sub-divided into two groups (n=9 and n=17) this further reduces the strength of any findings reported thus the reader cannot be confident that the findings are nothing more than spurious.

Response 3: Thank you for your great comment. As described in the Method section, patients with clinically inactive Crohn’s disease were enrolled in this study. However, the number of patients was small because we could collect in a certain period of time at a single medical institution. As we mentioned above, we recognize the small number of enrolled patients as the limitation of study, therefore, we added the following sentence in the part of Discussion section in the 308th line of page 12: we enrolled patients with clinically inactive Crohn’s disease at a single medical institution. There were no specific eligible criteria for the enrollment.

Point 4: No exclusion criteria were applied; why? This is surprising as there are several important factors that would need to be considered to ensure the study cohort are, as described, in clinical remission and also not taking medication/therapies that may impact the outcome measures of interest. It is also highly unusual for CD in particular where disease pathology is variable (i.e. differences in terms of site of disease, duration and course of medication etc.) making standardisation (using specific criteria) essential. Specifically, patients should be screened based on medication usage and those taking steroids, methotrexate and warfarin should be excluded. Those taking antibiotics should also be excluded due to their impact on the gut microbiota. Probiotic use (also potentially modulatory to the gut microbiota) was not excluded.

Response 4: Thank you for your great comment. Repeatedly again, we investigated the relationship between microbiota and vitamin K status based on the CD patients’ conditions in the real world, therefore, we did not decide the exact eligible criteria. Regarding the use of warfarin and MTX, there were no patients receiving these drugs in this study. We added the following sentence in the part of Results section in the 166th line of page 4: Regarding the use of warfarin and methotrexate, there were no patients receiving these drugs in this study.

Point 5: The rationale for use of the 4.5 ng/ml cut point for ucOC is not justified. The authors site one paper for use of this cut-point yet ucOC values are veriable even within apparently similar populations. Furthermore, reporting ucOC as a stand-alone measure of vitamin K status is of limited value. As recommended by Gundberg et al. (1998) it is not accurate to report just ucOC (as an absolute value) without considering total osteocalcin available (which is variable between people). Thus %ucOC should be expressed (considering both cOC and ucOC) when considering this biomarker of vitamin K status. This is not the case in the present study and is a major limitation.

Response 5: Thank you for your great comment. According to your suggestions, We added the following sentence in the part of Materials and Methods section in the 130th line of page 3: Regarding the cut off value for serum level of uc-OC, the cut-off value for serum level of uc-OC in osteoporosis is set at 4.5 ng/ml at many institutions in Japan based on the previous data in Japan [43].We added the following sentence in the part of Discussion section in the 313th line of page 12: using %uc-OC with more indicators may be ideal for the evaluation of vitamin K deficiency. However, in Japan, the measurement of uc-OC is approved for osteoporosis, while those of vitamin K and OC are not. Therefore, we evaluated vitamin K condition only by serum level of uc-OC, which is an available parameter in daily clinical practices in Japan.

43.Shiraki M.; Aoki C.; Yamazaki N.; Ito Y.; Tsugawa N.; Suhara Y.; Okano T. Clinical assessment of undercarboxylated osteocalcin measurement in serum using an electrochemiluminescence immunoassay: Establishments of cut-off value to determine vitamin K insufficiency in bone and to predict fracture leading to clinical use of vitamin K2. Jpn J Med Pharm Sci 2007, 57, 537-546 (in Japanese).

Point 6: Triglycerides are also not included in the variables reported – this has an impact on vitamin K absorptive capacity and may lead to differences in ucOC levels between patients.

Response 6: Thank you for your great comment. According to your comment, we added the Items such as triglycerides in the part of Materials and Methods section in the 142th line of page 4 and in the part of Results section in the Table 1 of page 5.

Point 7: The conclusion suggests this study proves cause and effect of gut microbiota variability on vitamin K status. This is not the case and it is premature to suggest this based on the limited evidence provided here.

Response 7: Thank you for your great comment. As you pointed out, there are many limitations in this study. Our conclusion might be exaggerated. Therefore, we changed the conclusions in the 334th line of page 12 from “To prevent intestinal inflammation and osteoporosis associated with CD in the future, therapeutic intervention with manipulation of gut microbiota and vit K might be considered even in patients with clinically inactive CD.” to “These data suggested the significance of investigating the gut microbiota even in patients with clinically inactive CD for improving patients’ vit K status.”

We changed the abstract in the 38th line of page 1 from “Taken together, to prevent intestinal inflammation and osteoporosis associated with Crohn’s disease in the future, therapeutic intervention with manipulation of gut microbiota and vitamin K might be considered even in patients with clinically inactive Crohn’s disease.” to “Taken together, these data suggested the significance of investigating the gut microbiota even in patients with clinically inactive CD for improving patients’ vit K status.”

Reviewer 2 Report

See my comments in enclosed draft PDF.

The authors shopuld be a little more careful and make a better discussion on their uc-OC finds and discuss that to current knowledge of uc-OC and osteogenesis. What levels have been seen in malnutrition, very old patients with immobilisation and fractures, dementia etc (insufficient food intake), osteoporois research.

Laboratory methods description of PIVKA and uc-OC are needed.

Sampling, sample processing?

Discussion on vitamin K producing bacteriae in colon contra production/absorption in colon and in small intestine.

I think this is an interesting study, but also complex because there are so many issues on vitamin K1/K2 in nutrition, adequate supply, absorption, chylomicrone production and transport, liver storage and retransport to extrahepatic tissues with effects on bone, brain, arteries and more.... Vitamin K2 supplementation  and atherosclerosis with MGP is also a tricky area and it has only been shown to be significant in diabetic patients needing dialysis.

Something on Menaquinone 7 which is the vitamin K2 to favour.

Author Response

Response to Reviewer 2 Comments

Point 1: The authors shopuld be a little more careful and make a better discussion on their uc-OC finds and discuss that to current knowledge of uc-OC and osteogenesis. What levels have been seen in malnutrition, very old patients with immobilisation and fractures, dementia etc (insufficient food intake), osteoporois research.

Response 1: Thank you for your comment. As for the nutritional condition, enrolled inactive CD patients in this study had maintained clinical remission with keeping good nutritional status because laboratory data showed the mean serum level of albumin were 4.0 g/dL in vit K-normal group and 3.9 g/dL in vit K-deficient group. Of note, our data strongly indicates the increase of uc-OC even in CD patients with good nutritional condition, which is an important and informative result for physicians.Therefore, we added the following sentence in the part of Introduction section in the 97th line of page 3 as follows: The increase of serum uc-OC indicates vit K deficiency. Dietary vit K intake for healthy women affects serum levels of uc-OC [32]. The serum levels of uc-OC negatively correlated with the vit K intake in anorexia nervosa patients showing bone loss [33]. Elevated serum level of uc-OC indicates the increased risks of hip fractures in elderly women [34,35]. It was reported that the serum levels of uc-OC were more strongly related to ultrasound transmission speed than to femoral neck density [36]. An observational study on adult patients with CD, demonstrated that the serum levels of uc-OC positively correlated with bone turnover speed [37] and to negative correlate with BMD in the lumbar spine [23].We added the following word in the Results section in the 165th line of page 4: albumin. We added the following sentence in the Discussion section in the 278th line of page 11: As for the nutritional condition, enrolled inactive CD patients in this study had maintained clinical remission with keeping good nutritional status because laboratory data showed the mean serum level of albumin were 4.0 g/dL in vit K-normal group and 3.9 g/dL in vit K-deficient group. Of note, our data strongly indicates the increase of uc-OC even in CD patients with good nutritional condition, which is an important and informative result for physicians.

Point 2: Laboratory methods description of PIVKA and uc-OC are needed.

Response 2: Thank you for your comment. According to your comment, we added the following sentence in the Materials and Methods section in the 145th line of page 4 PIVKA-II and uc-OC levels were measured by electrochemiluminescent immunoassay (ECLIA).

Point 3: Sampling, sample processing?

Response 3: Thank you for your comment. As described in the Method section. We added the following sentence in the Discussion section in the 308th line of page 12: we enrolled patients with clinically inactive Crohn’s disease at a single medical institution. There were no specific eligible criteria for the enrollment.

Point 4: Discussion on vitamin K producing bacteriae in colon contra production/absorption in colon and in small intestine.

Response 4: Thank you for your great comment. According to your comment, we added the sentence in the Introduction section in the 74th line of page 2 as follows: Vit K gets absorbed by the enterocytes of the small intestine and is dependent on bile, pancreatic enzymes and the dietary fat content. Vit K2 in human intestines is related to the composition of the intestinal microflora but is reported to be highly variable [19]. Also, while vit K is absorbed in the small intestine through a mechanism that requires bile acid salts, most is produced in the colon where there are no bile acid salts. The percentage of vit K synthesized by intestinal bacterial is much lower than excepted previously, although the exact proportion of vit K production/absorption in colon and small intestine remain unclear [20]. However, due to the difference of bioavailability, bioactivity and several unique functions of vit K2 [3,21], it is considered that even small amounts of vit K2 derived from intestinal bacteria can have a significant impact on health.

Point 5: I think this is an interesting study, but also complex because there are so many issues on vitamin K1/K2 in nutrition, adequate supply, absorption, chylomicrone production and transport, liver storage and retransport to extrahepatic tissues with effects on bone, brain, arteries and more.... Vitamin K2 supplementation  and atherosclerosis with MGP is also a tricky area and it has only been shown to be significant in diabetic patients needing dialysis.

Response 5: Thank you for your great comment. According to your suggestions, we added the following sentence in the Introduction section in the 58th line of page 2: Also, vit K is reported to be related to various organs and diseases such as the chronic kidney disease, some types of cancer, liver disease, immunity functions, neurological disorders, obesity and so on. Vit K1 is preferentially retained in the liver to aid the carboxylation of coagulation factors [9]. In contrast, vit K2 is redistributed into the circulation and is used for extrahepatic tissues such as bones and the vascular system [9]. The mechanism of clinical outcomes can be complicated because the work of vit K varies widely.

Point 6: Something on Menaquinone 7 which is the vitamin K2 to favour.

Response 6: Thank you for your great comment. According to your comment, we added the following sentence in the Introduction section in the 48th line of page 2: Vit K2 is classified by the number of isoprene units in the side chain. MK-4 is relatively abundant in chicken meat and egg. Natto, which is a Japanese fermented soybean-based product, is rich in MK-7, also contains MK-8. Several types of cheese contain MK-8 and MK-9 [3]. The vit K2 synthesized by intestinal bacteria are mainly MK-10 and MK-11, but small amounts of MK-7, MK-8, MK-9 and MK-12 are also synthesized [4].

Reviewer 3 Report

>There are 17 types of Vitamin K dependent Protein and reading the manuscript seem existing only two VKPDs (MGP and BGP)
>page 2, line 53: The authors wote very wrong (bad) sentence: Vitamin K is rare in healthy individuals
>manuscripts cited are very old and current knowledge is changed
I>n the study design they didn't consider warfarin use as cause vitamin K deficiency
>They didn't consider relevant bone biomarker in a correct evaluation of the osteoporosis: 25(OH)D ( osteocalcin is under the control of vitamin D, which in turn regulates BGP gene expression), ALP, PTH,CTX. Therefore statistical analysis is inadequate and their results can't be considered reliable

Author Response

Response to Reviewer 3 Comments

Point 1: There are 17 types of Vitamin K dependent Protein and reading the manuscript seem existing only two VKPDs (MGP and BGP)

Response 1: Thank you for your great comment. According to your comment, we added the following sentence in the Introduction section in the 54th line of page 2: In addition, there are Vit K-dependent proteins that are carboxylated by vit K in places like periosteum and periodontal membrane (periostin), brain and vascular endothelial tissues (growth arrest specific-6), liver (transthyretin), cornea (β-Ig-H3), biomembrane (PRGP-1, 2; TMGP-3, 4) and so on.

Point 2: page 2, line 53: The authors wote very wrong (bad) sentence: Vitamin K is rare in healthy individuals

Response 2: Thank you for your comment. In the text, the sentence is written not as: “Vit K is rare in healthy individuals”, but as: “Vit K deficiency is rare in healthy individuals”. Can you please check once again?

Point 3: manuscripts cited are very old and current knowledge is changed

Response 3: Thank you for your comment. According to your comment, we added the current references.

Point 4: In the study design they didn't consider warfarin use as cause vitamin K deficiency

Response 4: Thank you for your great comment. Regarding the use of warfarin and MTX, there were no patients receiving these drugs in this study. We added the following sentence in the part of Results section in the 166th line of page 4: Regarding the use of warfarin and methotrexate, there were no patients receiving these drugs in this study.

Point 5: They didn't consider relevant bone biomarker in a correct evaluation of the osteoporosis: 25(OH)D ( osteocalcin is under the control of vitamin D, which in turn regulates BGP gene expression), ALP, PTH,CTX. Therefore statistical analysis is inadequate and their results can't be considered reliable

Response 5: Thank you for your great comment. The relationship of uc-OC with vit D and bone markers has been considered in a previous report, so this time we focused on vit K, which is closely related to intestinal bacteria. In addition, we mainly considered the examination parameters whose measurements are often performed in daily clinical practices.

According to your comment, we added intact-PTH in the Materials and Methods section in the 142th line of page 4 and in the results Introduction section in the Table 1 of page 5. we added the following sentence in the Discussion section in the 321th line of page 12: But, the relationship of uc-OC with vit D and bone markers has been considered in a previous report [10], so this time we focused on vit K, which is closely related to intestinal bacteria.

Round  2

Reviewer 1 Report

Despite the fact that the authors have tried to deal with the comments received - the fact remains that this study is not designed correctly and there are major limitations which preclude the authors from substantiating their findings.

Detail on how patients were selected/recruited is still lacking and the fact that no inclusion/exclusion criteria were applied is a major flaw, as there are several important variables which should have been considered which can have a major impact on their findings (as outlined in my first review). The lack of clarity on the patient cohort makes it difficult to determine the specific patient profile being studied and many variables (incl. antibiotic, probiotic and specific drug use, dietary intake) could have a direct impact on the outcome measures of interest. Furthermore the authors only use serum albumin as a measure of nutritional status which is not robust. They also suggest the reporting of absolute values of ucOC is clinically relevant in Japan however in research this is not the case. There was nothing to preclude the authors for measuring total osteocalcin and expressing ucOC as a percentage of the total which is the accepted method in research.

There is no justification for the small sample size and due to sub-categorisation this study is extremely underpowered to substantiate any of the findings reported. Thus this major limitation is not one which can be dealt with without increasing the sample size and properly profiling the patient cohort. 

Author Response

Response to Reviewer 1 Comments (round 2)

Point 1: Despite the fact that the authors have tried to deal with the comments received - the fact remains that this study is not designed correctly and there are major limitations which preclude the authors from substantiating their findings.

Response 1: Thank you for your great comment. Again, we strongly consider that this study is not clinical trial but observational study, and this study was investigated in the real world.

Point 2: Detail on how patients were selected/recruited is still lacking and the fact that no inclusion/exclusion criteria were applied is a major flaw, as there are several important variables which should have been considered which can have a major impact on their findings (as outlined in my first review). The lack of clarity on the patient cohort makes it difficult to determine the specific patient profile being studied and many variables (incl. antibiotic, probiotic and specific drug use, dietary intake) could have a direct impact on the outcome measures of interest.

Response 2: Thank you for your great comment. Eligibility criteria and exclusion criteria were not listed because they were not items set in consideration of the relationship with vitamin K. According to your comment, we changed the sentence in Study participants in the 118th line of page 3 to the following sentence: Eligibility criteria were outpatients with CD treated at Kyoto University Hospital, who agreed to participate in this study between August and December 2015. CD was diagnosed based on clinical diagnostic criteria by symptoms, radiological findings, endoscopic findings, and histological findings. Exclusion criteria were (1) Patients with clinically active disease, (2) Patients that are pregnant or are likely to be pregnant, (3) Patients who did not agree with the epidemiological study, and (4) Other cases deemed inappropriate by the attending physician or the conducting physician. There were 26 patients with CD from whom we obtained consent and from whom we were able to collect feces samples.

Point 3: Furthermore the authors only use serum albumin as a measure of nutritional status which is not robust.

Response 3: Thank you for your comment. According to your comment, we added the following sentence in the Discussion section in the 180th line of page 4: indicator of nutritional status (such as albumin, total cholesterol, triglyceride, cholinesterase, folic acid, vitamin B12, and homocysteine). We added the items in the Results section in the Table 1 of page 5. We changed the sentence in the discussion section in the 295th line of page 12: from ‘’because laboratory data showed the mean serum level of albumin were 4.0 g/dL in vit K-normal group and 3.9 g/dL in vit K-deficient group’’ to ‘’in laboratory data’’

Point 4: They also suggest the reporting of absolute values of ucOC is clinically relevant in Japan however in research this is not the case. There was nothing to preclude the authors for measuring total osteocalcin and expressing ucOC as a percentage of the total which is the accepted method in research.

Response 4: Thank you for your great comment. Main focus of our study was on the evaluation of tools that can be used in daily clinical practice. It is difficult to use OC for daily clinical practice because the measurement is not covered by Japan government insurance. We added the items in the Results section in the 329th line of page 13: covered by Japan government insurance.

Point 5: There is no justification for the small sample size and due to sub-categorisation this study is extremely underpowered to substantiate any of the findings reported. Thus this major limitation is not one which can be dealt with without increasing the sample size and properly profiling the patient cohort.

Response 5: Thank you for your great comment. As we replied previously, we recognize the small number of enrolled patients as the limitation of study. We consider that multicenter examination is necessary to increase the number of cases in future. In addition, we consider that it is necessary to make it a prospective test, to set exclusion criteria for items related to vitamin K, and to make items related to vitamin K between groups consistent. However, when conducting large clinical trials such as multicenter prospective trials, it is not ethically acceptable without the expected results and underlying data from past studies. At first, we examine whether there is a tendency in cross-sectional research in daily clinical practice like this study, and we strongly believe that we should proceed to clinical trials based on the results. Although the number of cases was small, to the best of our knowledge, this study was the first to demonstrate the association between gut microbiota and uc-OC, which is an alternative indicator of vitamin K deficiency, in patients with CD.

Reviewer 3 Report

Now manuscript is suitable for publication

Author Response

Response to Reviewer 3 Comments (round 2)

Point 1: Now manuscript is suitable for publication

Response 1: We are extremely grateful for you for valuable comments to our manuscript.
